# Unsupervised Domain Adaptation through Shape Modeling for Medical Image Segmentation

**Yuan Yao**[1]     YAOYUAN2000@SJTU.EDU.CN

**Fengze Liu**[2]     LIUFZ13@GMAIL.COM

**Zongwei Zhou**[2]     ZZHOU82@JH.EDU

**Yan Wang**[3]     YWANG@CEE.ECNU.EDU.CN

**Wei Shen**[1]     SHENWEI1231@GMAIL.COM

**Alan Yuille**[2]     ALAN.L.YUILLE@GMAIL.COM

**Yongyi Lu**[2]     YYLU1989@GMAIL.COM

[1] *Shanghai Jiao Tong University*

[2] *Johns Hopkins University*

[3] *Shanghai Key Laboratory of Multidimensional Information Processing, East China Normal University*

**Editors:** Under Review for MIDL 2022

## Abstract

Shape information is a strong and valuable prior in segmenting organs in medical images. However, most current deep learning based segmentation algorithms have not taken shape information into consideration, which can lead to bias towards texture. We aim at modeling shape explicitly and using it to help medical image segmentation. Previous methods proposed Variational Autoencoder (VAE) based models to learn the distribution of shape for a particular organ and used it to automatically evaluate the quality of a segmentation prediction by fitting it into the learned shape distribution. Based on which we aim at incorporating VAE into current segmentation pipelines. Specifically, we propose a new unsupervised domain adaptation pipeline based on a pseudo loss and a VAE reconstruction loss under a teacher-student learning paradigm. Both losses are optimized simultaneously and, in return, boost the segmentation task performance. Extensive experiments on three public Pancreas segmentation datasets as well as two in-house Pancreas segmentation datasets show consistent improvements with at least 2.8 points gain in the Dice score, demonstrating the effectiveness of our method in challenging unsupervised domain adaptation scenarios for medical image segmentation. We hope this work will advance shape analysis and geometric learning in medical imaging.

**Keywords:** Segmentation, Unsupervised Domain Adaptation, 3D Shape Modeling.

## 1. Introduction

Semantic segmentation of anatomical organs is a challenging task in clinical research. One major problem in practice is that the preferred modality and scanning protocol that different hospitals adopt can vary significantly. Different CT machines and protocols result in different spacing, slice thickness of the scans, and variance of intensity, textures of the organs. Therefore, models trained on a specific source domain from a certain hospital can often decrease performance if directly applied to data obtained from other hospitals without fine-tuning towards the target data. For example, training a 3D U-Net (Özgün Çiçek

et al., 2016) on a public NIH Pancreas dataset[1] and directly testing it on another public MSD Pancreas dataset (Simpson et al., 2019) yields 15.1% performance drop (from 0.829 to 0.678) in terms of Dice score. Moreover, medical image segmentation requires large-scale annotated data to achieve good performance, which is available in the labeled source domain but not in the unknown target domains. Thus, supervised fine-tuning on the target domain is not feasible. Owing to these observations, we seek to answer the following critical question in this paper - Can we improve domain adaptation for medical image segmentation tasks without target labels?

We hereby introduce an unsupervised domain adaptation (UDA) framework to take into account intrinsic shape statistics into standard medical image segmentation models (e.g., 3D U-Net). The intuition behind our work is that the shape representation learned from the source domain is beneficial for the segmentation task on the target domain as different datasets should share the same representation of the 3D anatomy if they are from the same organ (e.g., Pancreas), albeit the change of textures caused by different scanning machines, protocols, phases, etc. In addition, Liu et al. (2019) proved that Variational Autoencoder (VAE) based models can learn the distribution of shape for a certain organ and can be used to evaluate the quality of a segmentation prediction by fitting it into the learned shape distribution. Inspired by this, we propose a new UDA pipeline based on a dual-loss function under a teacher-student learning paradigm. On the source domain, the intrinsic organ shape statistics are captured via a pre-trained VAE, apart from a trained segmentation network, which will serve as the teacher network later. During domain adaptation, the target segmentation network (student network) is updated with the guidance of the trained segmentation network (teacher network) as well as a VAE reconstruction loss. Unlike traditional teacher-student approaches, which impose feature-level consistency between two networks, we further introduce a pseudo-label loss in pixel-level that favors dense prediction tasks. Both VAE reconstruction loss and pseudo loss are optimized simultaneously and, in return, boost the segmentation performance. We conducted experiments on three public Pancreas CT datasets as well as one in-house Pancreas CT dataset. Extensive results showed that our segmentation with explicit VAE shape modeling outperformed other domain adaptation methods with or without shape priors.

## 2. Related Work

Unsupervised domain adaptation exploits the labeled source data and unlabeled target data. The mainstream methods seek to reduce the domain discrepancy of images or features (Wilson and Cook, 2020; Toldo et al., 2020). For adaptation at the image level, image-to-image translation—either converting images from source to target domain (Taigman et al., 2016; Bousmalis et al., 2017) or learning a joint distribution (Liu and Tuzel, 2016; Sankaranarayanan et al., 2018)—can be accomplished by a conditional GAN. For adaptation at the feature level, the methods like adversarial training (Ganin and Lempitsky, 2015; Tzeng et al., 2017) and explicit domain discrepancy measures (Tzeng et al., 2014; Long et al., 2015, 2016, 2017) can build features that are invariant across the domains. In the context of domain adaptive semantic segmentation tasks, AdaptSegnet (Tsai et al., 2018) attempts to align the distribution at the output level, whereas SIFA (Chen et al., 2020)

---

1. https://wiki.cancerimagingarchive.net/display/Public/Pancreas-CT

suggests adapting the distribution at both image-level and feature-level. A common trend among these methods is to *explicitly* align the distributions across the domains. In addition, self-supervised methods—integrated with mean teacher (Nguyen et al., 2021a; Deng et al., 2021; Nguyen et al., 2021b) and pseudo labeling (Gu et al., 2020; Zheng and Yang, 2021)— encourage to *implicitly* align the distributions and have recently achieved compelling results on multiple tasks. Although tremendous progress has been made in this field, most existing methods have not incorporated the object shape information into the domain transfer.

Apart from UDA, some works incorporate the shape prior into the segmentation pipeline to optimize the results. Projective convolution network (Kalogerakis et al., 2017) focuses on the view-based shape representations. Shape denoising network (He et al., 2021) suggests to extract a self-taught shape representation by leveraging weak labels, and then utilize this cue for shape refinement. However, in these task-specific shape models, domain shift is not delicately addressed, thus it hampers its usage on domain adaptation tasks. PCA (Milletari et al., 2017) method constructs shape models, but it requires further annotation for the key points. In abdominal imaging, the shape of most organs is naturally consistent under various domain shifts (e.g., imaging protocols, scanners, contrast enhancements, and human poses). Unlike all existing works, we are among the first to propose an end-to-end shape model inferring strong learning objective for abdominal organ segmentation without extra annotation across the domains.

## 3. Method

We first formulate the unsupervised domain adaptation (UDA) for segmentation task. In UDA setting, there are two segmentation datasets, the labeled one as the source domain and the unlabeled one as the target domain. We denote a segmentation dataset $\{x_i, y_i\}_{i=1}^n$ where $x_i$ is an image and $y_i$ is the corresponding pixel-wise ground truth annotation of $x_i$. The source dataset can therefore be denoted as $\{x_i^s, y_i^s\}_{i=1}^N$, and the target dataset as $\{x_i^t, y_i^t\}_{i=1}^M$, where $y^t$ is unknown in the UDA task. The goal is to train a segmentation network $S$ to predict the labels $y^t$ on target images $x^t$.

A straightforward way to solve this problem is to train $S$ on the labeled source domain data, and directly predict labels on the target domain by $S(x^t)$. However, domain gap exists due to biases across different datasets, especially for medical imaging (such as bias and variability of computed tomography texture feature measurements across different clinical image acquisition settings). To narrow the domain gap between datasets, we explicitly introduce shape priors in the segmentation pipeline in observing the fact that unlike textures variance, shape context is relatively unchanged (e.g., the shape of the pancreas is consistent despite different clinical image acquisitions). Modeling shape on the source domain, we develop a Teacher-Student paradigm to finetune $S$ on the unlabeled target domain. Formally, we want to find a teacher model $L$ with the explicit modeling of shape, so that

$$L_\theta(S, x^t) = \mathcal{L}(S(x^t), y^t), \tag{1}$$

where $\theta$ is the parameters of model $L$ and $\mathcal{L}$ is a loss term to depict the similarity of segmentation results and ground truth. Liu et al. (2019) proved that Variational Autoencoder (VAE) based models can learn the distribution of shape for a certain organ, and can be used to evaluate the quality of a segmentation prediction by fitting it into the learned shape

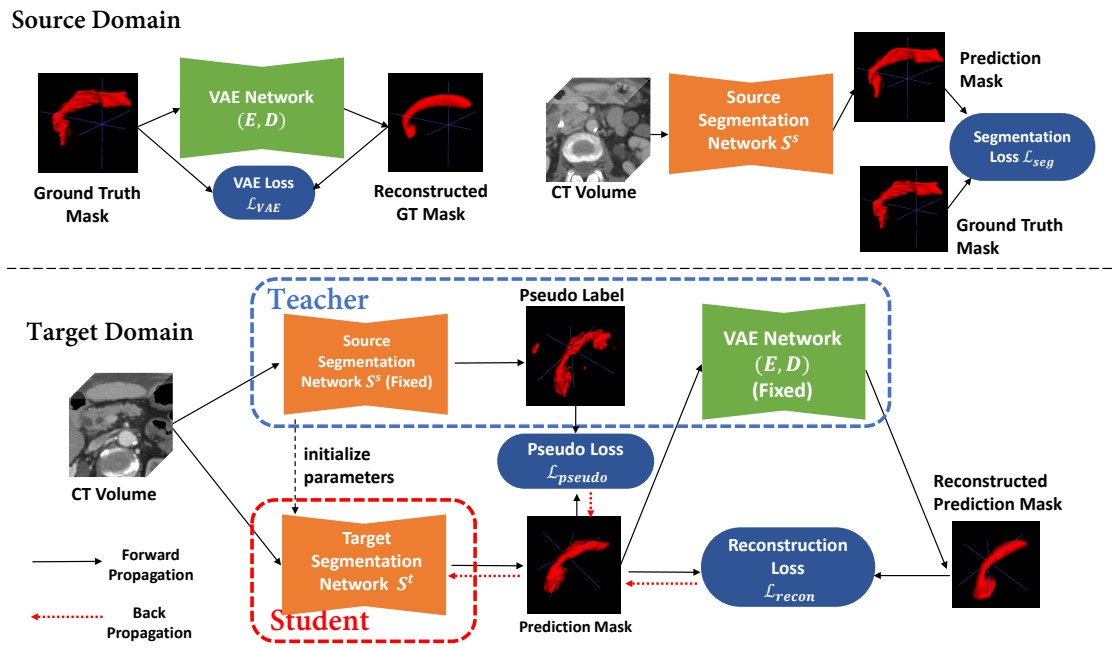

Figure 1: Proposed VAE based pipeline of unsupervised domain adaptation for medical segmentation.

distribution. In our work, we propose to incorporate the VAE based model into the teacher model to improve the segmentation results on the target domain.

The architecture of our proposed VAE-based UDA pipeline can be found in Figure 1. Our pipeline basically consists of two steps. In the first step, the VAE network is trained on the source domain with the objective to reconstruct the ground truth masks. The segmentation network is also trained on the source domain. In the second step, we first copy the parameters of source segmentation network to the target segmentation network as its initial weights, and fix the parameters of the VAE network. We then update the target segmentation network by jointly optimizing two losses, a pseudo-loss and a VAE reconstruction loss. These steps are described with more details in following sections.

### 3.1. Segmentation Network

In our proposed teacher-student paradigm, the segmentation network from source domain serves as a teacher model while the target segmentation network as a student model. The segmentation network we use in our work is 3D-UNet (Özgün Çiçek et al., 2016), a common and effective network architecture in medical image segmentation. Using Dice coefficient as the loss term, we formally define the segmentation loss as

$$\mathcal{L}_{seg} = -\frac{2\|S(x^s) \cdot y^s\|_1}{\|S(x^s)\|_1 + \|y^s\|_1}. \tag{2}$$

### 3.2. Modeling Shape with VAE

To model the shape in VAE feature space, we assume there is some distribution $P(y)$ for normal label $y$, and we train a VAE network on ground truth masks $\{y_i^s\}_i^N$ in the source domain. In the VAE network, we aim to find an estimation function $Q(z|y)$ that gives a distribution of latent vector $z$ that are likely to produce ground truth mask $y$. The objective to optimize in the VAE network is

$$\log P(y) - \mathcal{KL}[Q(z|y)\|P(z|y)] = \mathbb{E}_{z \sim Q}[\log P(y|z)] - \mathcal{KL}[Q(z|y)\|P(z)], \tag{3}$$

where $\mathcal{KL}$ is Kullback-Leibler divergence.

As is demonstrated in Liu et al. (2019), the right hand side of Equation (3) becomes a natural estimation of $\log P(y)$ during optimization of the VAE objective. Experiments show that term $\mathcal{KL}[Q(z|y)\|P(z)]$ is less related to the distribution of $y$, so we can take $\mathbb{E}_{z \sim Q}[\log P(y|z)]$ as an evaluation of shape.

Now we interpret the VAE objective. VAE network basically contains a encoder $E = (\mu, \Sigma)$ that estimates the mean and variance of the Gaussian distribution for latent vector $z$ and a decoder $D$ that reconstructs $z$. Here, we use Dice coefficient as the loss term, and choose Gaussian Distribution for the distribution of latent vector $z$. VAE network is trained on the source domain. The formal definition of the VAE training loss is

$$\mathcal{L}_{VAE} = -\mathbb{E}_{z \sim \mathcal{N}(\mu(y^s), \Sigma(y^s))} \left( \frac{2\|y^s \cdot D(z)\|_1}{\|y^s\|_1 + \|D(z)\|_1} \right) + \lambda_{\mathcal{KL}} \cdot \mathcal{KL}\left( \mathcal{N}(\mu(y^s), \Sigma(y^s)) \| \mathcal{N}(0,1) \right), \tag{4}$$

where $\lambda$ is a hyperparameter regulating the two terms.

The shape benchmark can be interpreted as the VAE reconstruction loss, which is Dice Loss between ground truth label and the label reconstructed by VAE. VAE reconstruction loss serves as part of the teacher model when finetuning the segmentation on the target domain. It can be mathematically written as

$$\mathcal{L}_{recon} = -\frac{2\|y^t \cdot D(\mu(y^t))\|_1}{\|y^t\|_1 + \|D(\mu(y^t))\|_1}, \tag{5}$$

where we simply estimate the latent variable using mean value $\mu(y^t)$.

### 3.3. VAE-based UDA Pipeline

In VAE-based UDA pipeline, the first step is to train a segmentation network $S^s$ using $\mathcal{L}_{seg}$ and a VAE network $(E, D)$ using $\mathcal{L}_{VAE}$ in the source domain. In the unlabeled target domain, the parameters of target segmentation network are copied from source segmentation network and finetuned with a teacher model. Recall that we want to find a teacher model $L_\theta$ with the explicit model of shape, so that

$$L_\theta(S^t, x^t) = \mathcal{L}(S^t(x^t), y^t). \tag{6}$$

The shape estimation loss $\mathcal{L}_{recon}$ is part of $L_\theta^t(S, x^t)$, aiming to introduce the shape prior learned from the source domain to reduce biases. Noticing that reconstruction loss $\mathcal{L}_{recon}$

only derives from the foreground masks, another loss term is required to prevent the segmentation network outputting results deviating the image.

Pseudo labels predicted by source segmentation network can serve as a constraint that directs reconstruction loss. The Dice Loss between the predictions from the source and target segmentation network is defined as pseudo loss, which can be formally written as

$$\mathcal{L}_{pseudo} = -\frac{2\|S^s(x^t) \cdot S^t(x^t)\|_1}{\|S^s(x^t)\|_1 + \|S^t(x^t)\|_1}. \tag{7}$$

The effect of pseudo loss lies in two aspects. First, pseudo loss punishes the segmentation results with low confidence. Second, it serves as an adversarial counterpart of reconstruction loss, avoiding a shortcut solution of target network, i.e., generating an identical predicted mask that has relatively small reconstruction loss.

Based on the reconstruction loss and the pseudo loss which have adversarial effects, the teacher model can be defined as

$$L_\theta(S^t, x^t) = \lambda_{recon} \cdot \mathcal{L}_{recon} + \mathcal{L}_{pseudo}, \tag{8}$$

where $\lambda_{recon}$ is a hyperparameter mediating the two losses, and $S^s, \mu, D$ defined in the two losses are fixed networks parameterized by $\theta$. Besides, based on the observation that $\mathcal{L}_{recon}$ can be an estimator of the segmentation quality, $\mathcal{L}_{pseudo}$ should take a smaller weight when $\mathcal{L}_{recon}$ is high. Thus, we proposed a technique of dynamic hyperparameter for $\lambda_{VAE}$ in actual training. Specifically, the value of $\lambda_{recon}$ changes with regard to $\mathcal{L}_{recon}$.

### 3.4. Test-time Training

VAE contains the shape prior not only benefits the training process, but also testing. Thus, a test-time training method is proposed to better exploit the effect of VAE network. For each test image $x_{test}^t$, we adopt the loss $L_\theta(S^t, x^t)$ given by the teacher model and train the target segmentation network $S^t$ for 1 iteration to get $\dot{S}^t$. The final segmentation result $y_{test}^t$ for test image $x_{test}^t$ is then given by $y_{test}^t = \dot{S}^t(x_{test}^t)$. After each prediction $\dot{S}^t$ is discarded.

## 4. Experiments

### 4.1. Datasets and Metrics

In the experiment part, we used three public pancreas CT datasets (NIH, MSD, Synapse) and two in-house pancreas CT datasets (IV contrast and oral contrast). NIH serves as the source domain in our experiment, where we trained the teacher networks (segmentation network and VAE network). The other three datasets serve as the target domain data for testing. We use the mean Dice score as the evaluation metric for segmentation results. The descriptions for the datasets can be found in Appendix D.

### 4.2. Ablation Studies

We conduct ablation studies on the transfer from the NIH dataset to the MSD dataset as the MSD dataset has a sufficient number of images. Several proposed components of our

Table 1: Ablation study of key components in our proposed VAE-based pipeline.

| Pseudo Loss | Reconstruction Loss | Dynamic Hyperparameter | Test-time Training | Dice |
|:---:|:---:|:---:|:---:|:---:|
| - | - | - | - | 0.6777 |
| ✓ | - | - | - | 0.7068 |
| - | ✓ | - | - | Not work |
| ✓ | ✓ | - | - | 0.7484 |
| ✓ | ✓ | ✓ | - | 0.7529 |
| ✓ | ✓ | ✓ | ✓ | **0.7574** |

Figure 2: The 2D and 3D visualization for the results on a MSD test case.

network structure are evaluated, including pseudo loss $\mathcal{L}_{pseudo}$, reconstruction loss $\mathcal{L}_{recon}$, dynamic hyperparameter and test-time training.

Table 1 is a summary of the ablation study of components of our network. The row "Baseline" demonstrates the result of the direct test of 3D U-Net. Specifically, the segmentation network is trained on the NIH training set and directly tested on the MSD validation set. This should be the lower bound for our network architecture.

To validate the effectiveness of our network loss, we fine-tune the network on MSD training set using only $\mathcal{L}_{pseudo}$, which yields 0.029 Dice score improvement. Then we incorporate the VAE reconstruction loss with the pseudo loss. This shows another 0.042 Dice score improvement. When exploring the necessity of pseudo loss, we find that it is not workable only training with reconstruction loss. The segmentation will deviate from the background image in this case, as VAE only deals with the distribution of foreground masks. The last two rows of this table show the effectiveness of several learning techniques. In the Dynamic Hyper-parameter experiment, we set different choose different $\lambda_{VAE}$ depending on $\mathcal{L}_{recon}$. The hyperparameter experiment shows a 0.005 improvement on the dice score. We further implemented the test-time fine-tuning technique and gained another 0.005 improvements.

### 4.3. Comparison with other UDA methods

Extensive experiments on three target domains were performed to compare our VAE-based model with baseline models for segmentation in domain adaptation. All the methods are tested on the same split with the sa me data preprocessing method. The implementation details can be found in Appendix B. The experiment results are shown in Table 2.

Table 2: Performance comparison of VAE-based pipeline with other UDA segmentation methods. The segmentation results are evaluated with mean Dice score. Domain gap is calculated by the dice gap between a certain method and the upper bound.

| Method | MSD | | Synapse | | In-house IV | | In-house Oral | |
|---|---|---|---|---|---|---|---|---|
| | Dice | Gap | Dice | Gap | Dice | Gap | Dice | Gap |
| Direct Test | 0.6777 | 0.1283 | 0.7452 | 0.0509 | 0.7983 | 0.0657 | 0.7019 | 0.1208 |
| Pseudo Label | 0.7068 | 0.0992 | 0.7769 | 0.0192 | 0.8139 | 0.0501 | 0.7378 | 0.0849 |
| Discriminator | 0.7176 | 0.0884 | 0.7817 | 0.0144 | 0.8127 | 0.0513 | 0.7296 | 0.0931 |
| SIFA | 0.6605 | - | 0.7456 | - | 0.7758 | - | 0.6910 | - |
| VAE pipeline (ours) | **0.7574** | **0.0486** | **0.7869** | **0.0092** | **0.8264** | **0.0376** | **0.7453** | **0.0774** |
| *Upper bound* | *0.8060* | - | *0.7961* | - | *0.8640* | - | *0.8227* | - |

Among these baseline methods, a direct test is the lower bound that we train a 3D U-Net on the source domain and directly test it on the target domain. Adopting pseudo labels in the target domain to predict the ground truth labels is an intuitive method in unsupervised domain adaptation. Beyond pseudo labels, the discriminator is a generative-model based network that focuses on label alignment by explicitly applying a discriminator. (See Appendix E.) SIFA is a 2D GAN-based network dealing with feature-level and image-level alignment simultaneously. Besides, we also tested the upper bound of domain adaptation by fine-tuning 3D U-Net on the target domain using labeled training data. We calculate the domain gap by the difference between upper bound dice and dice of a certain method. The domain gap of SIFA is not calculated because its backbone is not 3D U-Net.

Figure 2 presents an example of segmentation results on MSD target dataset for each method. Examples of other target datasets can be found in Appendix F. Due to the domain gap, a direct test with segmentation trained on the source domain may lead to results with noise on the target domain. The pseudo label can reduce noises, but the segmentation results are still not perfect since the pseudo label itself suffers from bias. Though discriminator can incorporate distribution prior information into the UDA pipeline to some extend, the discriminator loss is not as effective as VAE reconstruction loss in maintaining a specific distribution in the global context. SIFA does not perform well in our experiment, for it only deals with 2D figure and may fail on some slices. By incorporating VAE in the UDA pipeline, our model conforms with the pancreas' shape better than all other methods. This validates VAE's power of incorporating shape prior.

## 5. Conclusions

We proposed an unsupervised domain adaptation method to generalize 3D segmentation models to medical images collected from different scanners and/or protocols (domains). Our method is inspired by the fact that organs usually show consistent shape, i.e., contours, between most modalities and protocols, while texture and intensity can vary significantly. We incorporated shape features into the segmentation network via variational autoencoder (VAE) and utilized two losses, i.e., a VAE reconstruction loss and a pseudo loss, to guarantee its transferability to multiple domains. Experimental results on four target datasets demonstrate the superiority of our method.

## Acknowledgement

This work was supported by the Fundamental Research Funds for the Central Universities.

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

## Appendix A. Why it Works

To clearly demonstrate why our method works, we focus on the key losses in our proposed teacher model. Taking the validation data from MSD dataset as an example, Figure 3 demonstrates the distribution of data points with regard to their $\mathcal{L}_{recon}$ and $\mathcal{L}_{pseudo}$. The results of pseudo labels, ground truth masks and our predicted masks are demonstrated from left to right.

Note that we initialize the parameters using weights of source segmentation network, the pseudo labels are actually a starting point for our domain adaptation. Our goal is to make its distributions of the two losses as similar as that of the ground truth. With the statistics, we clearly see that our VAE-based pipeline managed to do so.

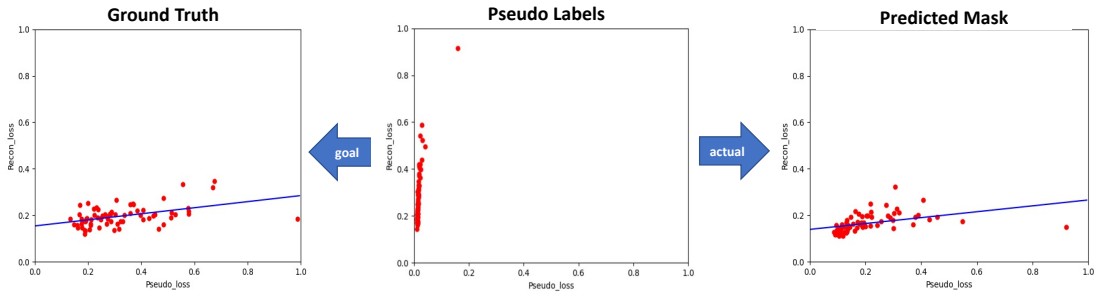

Figure 3: An analysis figure about the distributions of ground truth, pseudo label and predicted mask for data in MSD validation set. The y-axis represents $\mathcal{L}_{recon}$ and the x-axis represents $\mathcal{L}_{pseudo}$

.

## Appendix B. Implementation Details

As the voxel size varies among different data, we first preprocessed the training and validation data to the same voxel size of $1mm \times 1mm \times 1mm$. We also adopted a cube bounding box that is sufficient to hold the annotation mask and cropped the images and ground-truth masks on both the source domain and target domain.

Our 3D UNet backbone consists of 5 down-sampling blocks and 5 up-sampling blocks with skip connections. Each down-sampling block of input channel $c_{in}$ and output channel $c_{out}$ contains one 3D Conv layer of input and output channel $c_{in}$, one 3D Conv layer of input channel $c_{in}$ and output channel $c_{out}$, and two 3D Conv layers of input and output channel $c_{out}$. We take batch normalization and ReLU activation after the last three layers of the network. The up-sampling block is similar to the down-sampling block, except that it replaces the first 3D Conv layer with the 3D Conv Transpose layer. The number of channels we take for 3D U-Net is $8, 16, 32, 64, 128, 256$. A Softmax layer is applied at the final step.

The structure of the VAE network includes an encoder and a decoder. The encoder network contains five down-sampling blocks and two paralleled fully-connected layers for

Table 3: Training time (source domain)

| 3DUNet | VAE |
|---|---|
| 10 hours | 19 hours |

Table 4: Training time (target domain MSD)

| VAE pipeline (finetune) | 3DUNet from scratch |
|---|---|
| 3 hours | 14 hours |

generating the mean and variance. The decoder network contains five up-sampling blocks. A Softmax layer is applied at the final step. The number of dimensions we choose here for the bottleneck is 16384. We take $\lambda_{\mathcal{KL}}$ to be $2e-5$.

For the choice dynamic hyperparameter $\lambda_{VAE}$, we set three thresholds 0.15, 0.225, 0.3, dividing $\mathcal{L}_{recon}$ into four sections. We take $\lambda_{VAE}$ by multiplication with a factor $\gamma$ from $0.6, 1.2, 2.0, 3.0$ respectively. $\lambda_{VAE}$ is then given by $\gamma \cdot \hat{\lambda}_{VAE}$, where $\hat{\lambda}_{VAE}$ is the hyperparameter chosen in the experiment. In our experiment, we choose $\hat{\lambda}_{VAE} = 0.1$ on Synapse, and $\hat{\lambda}_{VAE} = 1$ on other target domains. We choose a smaller $\hat{\lambda}_{VAE}$ on Synapse due to its limited size. Loss term $\mathcal{L}_{recon}$ for the dataset with limited size will suggest a stronger template, and thus causing the segmentation model to pay less attention to the information of CT images.

Our network is trained with the SGD optimizer. We fix the learning rate to $1e-2$. All the frameworks are built on PyTorch. We train the source segmentation network and the VAE network for about $300,000$ iterations. We fine-tune the segmentation network on the target domain for about $15,000$ iterations. All experiments are trained and evaluated on a GPU server with four Nvidia TITAN Xp cards. The running time (approximately to convergence) can be found in Table 3 and Table 4.

## Appendix C. Ablation Study on the Adversial Effect of $\mathcal{L}_{recon}$ and $\mathcal{L}_{pseudo}$

Table 5: Ablation studies on weights of VAE reconstruction loss on MSD validation set.

| $\lambda_{VAE}$ | Dice |
|---|---|
| 0.0 | 0.7068 |
| 0.1 | 0.7131 |
| 0.2 | 0.7274 |
| 0.5 | 0.7367 |
| 1.0 | 0.7484 |
| 2.0 | 0.7361 |

When studying the effectiveness of VAE, we also experimented the adversial effects of pseudo loss and reconstruction loss. This is done by experiments of different $\lambda_{VAE}$. The experiments are conducted on the domain transfer from NIH to MSD, without the

techniques of dynamic hyper-parameter and test-time finetuning. Results can be found in Table 5.

## Appendix D. Datasets and Preprocessing

Here we provide descriptions about the datasets we take in our experiment.

- **NIH** Pancreas CT contains 82 abdominal contrast enhanced 3D CT scans. The CT scans have resolutions of $512 \times 512$ pixels with varying pixel sizes and slice thickness between $1.5 \sim 2.5$mm, acquired on Philips and Siemens MDCT scanners. The dataset is randomly splitted into a training set of 61 training cases and 21 testing cases.

- **MSD** contains 420 portal-venous phase 3D CT scans (282 Training + 139 Testing), having labels of pancreas and tumor. The CT scans have resolutions of $512 \times 512 \times l$ pixels. We merge the pancreas and tumor labels together as pancreas in our task. As we do not know the annotation on the test data, we randomly split the training set into our training set of 210 cases and testing set of 72 cases.

- **Synapse** contains 50 abdomen CT scans (30 Training + 20 Testing). Each CT volume consists of $85 \sim 198$ slices of $512 \times 512$ pixels, with a voxel spatial resolution of $([0.54 \sim 0.54] \times [0.98 \sim 0.98] \times [2.5 \sim 5.0])$mm$^3$. As we do not know the annotation on the test data, we randomly split the training set into our training set of 22 cases and testing set of 8 cases.

- **In-house IV Contrast Dataset.** Our in-house fully-labeled pancreas dataset includes 290 contrast-enhanced abdominal clinical CT images in the portal venous phase, in which we randomly choose 216/74 patients for training and testing. Each CT volume consists of $319 \sim 1051$ slices of $512 \times 512$ pixels, and have voxel spatial resolution of $([0.523 \sim 0.977] \times [0.523 \sim 0.977] \times 0.5)$mm$^3$, acquired on Siemens MDCT scanners.

- **In-house Oral Contrast Dataset.** Our in-house dataset also includes 91 oral contrast abdominal CT images in the portal venous phase, which are substantially different from the above four datasets in terms of the use of contrast agents, i.e., oral instead of intravenous (IV). We randomly choose 68/23 patients for training and testing. Each oral contrast CT volume consists of $512 \times 512 \times l$ pixels, and have voxel spatial resolution of $([0.523 \sim 0.977] \times [0.523 \sim 0.977] \times 0.5)$mm$^3$, acquired on Siemens MDCT scanners.

**Preprocessing.** We clip the intensities to a range of -200 to 400, then further normalize them into -1 to 1. Patch size of $128 \times 128 \times 128$ is used to sample training data. We apply augmentation of random intensity scaling of $0.85 \sim 1.15$, random rotation within 20 degrees and random translation within 5 voxels.

## Appendix E. More Descriptions of other UDA Competitors

In "Pseudo Label" and "Discriminator" baseline, we follow similar pipeline with our proposed VAE based pipeline. First train a segmentation network $S$ on the source domain, and

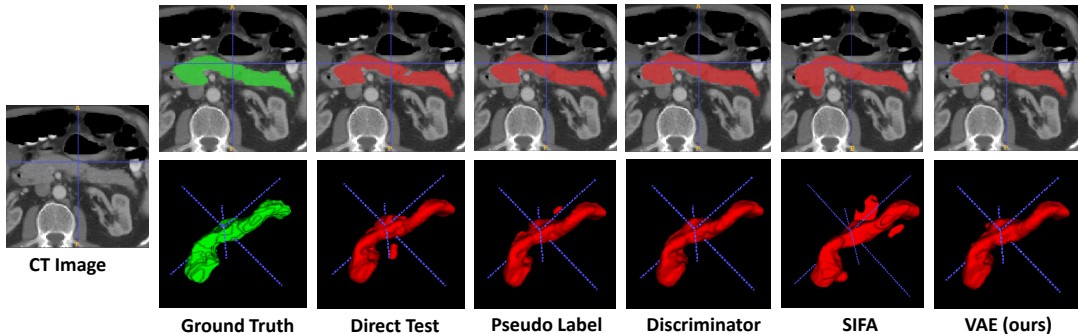

Figure 4: The 2D and 3D visualization for the results on a Synapse test case.

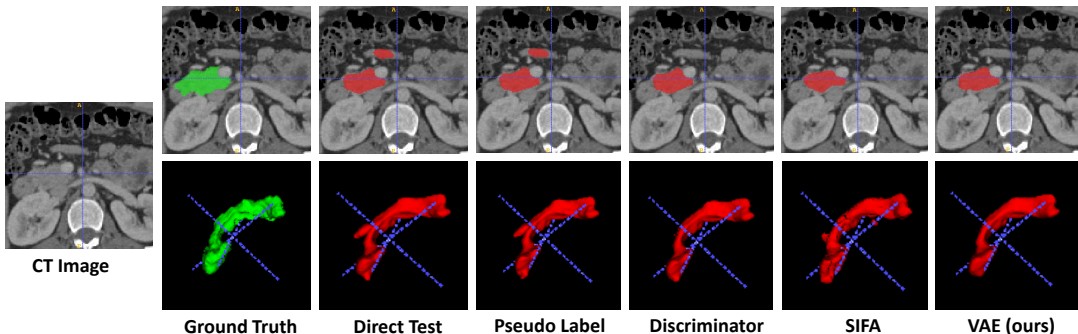

Figure 5: The 2D and 3D visualization for the results on an in-house IV contrast test case.

then finetune $S$ on the target domain under a teacher-student paradigm. In the "Pseudo Label" experiment, we only adopted pseudo loss on the target domain. In the "Discriminator" experiment, we further trained a discriminator on the source domain and implemented a discriminator loss on the target domain. The following is a more detailed description of the discriminator. We generated a number of masks (some of which with poor qualities) with several well-trained or poorly-trained segmentation networks in the source domain. We mix them with the ground truth masks, and train a discriminator to predict the dice score of these masks. The discriminator loss on the target domain is the score predicted by the discriminator.

## Appendix F. More Visualizations

Here are the visualizations on other target datasets. The visualization results of Synapse dataset can be found in Figure 4. The visualization results of in-house IV contrast dataset can be found in Figure 5. The visualization results of in-house oral contrast dataset can be found in Figure 6.

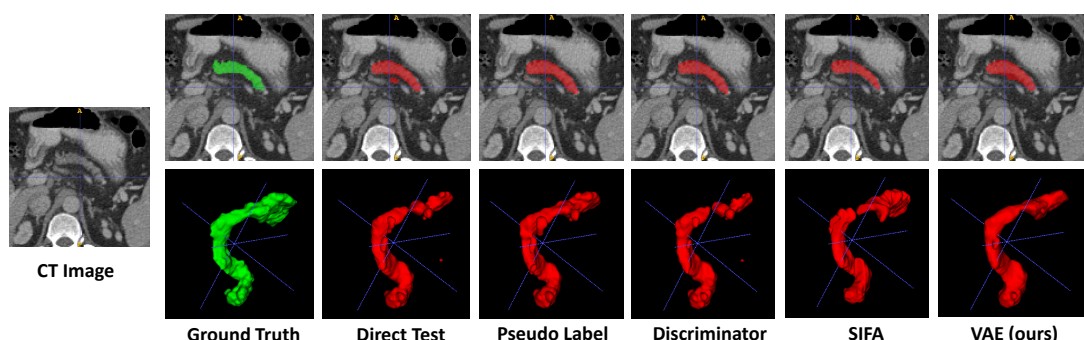

Figure 6: The 2D and 3D visualization for the results on an in-house oral contrast test case.

