# OpenReview forum: "Unsupervised Domain Adaptation through Shape Modeling for Medical Image Segmentation"
_MIDL.io/2022/Conference — MIDL 2022_

### Official Review · Reviewer_JpYa · 2022-01-14

**Confidence:** 4
**Preliminary Rating:** 3
**Recommendation:** Poster

**Summary:**

This paper proposes a solution to the unsupervised domain adaptation (UDA) problem for segmentation, by incorporating explicit shape information in the process. The proposed approach utilizes a combination of variational autoencoder (VAE) to learn a pancreas shape, 3D UNet for the segmentation and a teacher-student learning paradigm including a pseudo loss and a reconstruction loss for the DA. The method involves also test time training, which further improves the output quality. DA was tested on 3 different datasets, and an ablation study was conducted to show that each individual components have some benefits in the ultimate performance, which outperforms the baseline model.

**Strengths:**

- I would like to thank the authors for this really well written paper, easy to follow and fully comprehend. This was not easy to achieve since the approach involves multiple components.

- Most of the segmentation algorithms base their decisions mostly on texture related features. The main strength of this paper is the explicit introduction of shape information.

- The authors conducted an ablation study to investigate the effects of each component. They also do their best to explain why the approach works, with a very interesting graph explanatory of the adversarial behavior of pseudo and reconstruction loss.


**Weaknesses:**

- I think the biggest limitation of the work is with the related work section. Shape is not explicitly considered in many segmentation papers – true - however there is a large body of works which introduces shape to the segmentation pipeline, as a constrain, or as a guide. Some works that come to my mind are from Hoel Kervadec and coworkers. Kalogerakis et al., 3D shape segmentation with projective convolutional networks. F Milletari et al. Integrating statistical prior knowledge into convolutional neural networks, and many more recent papers can be found. I think that a more complete related work section is missing, and somehow it weakens the manuscript.

- The paper vaguely brings to memory the approach presented in the paper published at the last MIDL conference https://2021.midl.io/papers/a8 . Precisely the idea of learning a shape prior using a CNN, which is then used to correct the output of a different segmentation network. There are of course substantial differences, but I think it would be valuable to refer to that paper in the related work section, perhaps listing pros and cons of the approaches.

- The paper does not state any limitations with respect to the proposed approach. I think the work will benefit of some healthy self-criticism. An ablation study showing that each added component to the pipeline offers a performance improvement, does not make the work limitations-free.


**Deanonymize Review:**

no

**Detailed Comments:**

- It is not clear if the used pseudo label is a probability map, or the output is thresholded.

- In the reconstruction loss, was the DICE loss the only attempted loss, or have you tried other loss type such as CE, Focal loss, etc..?

- It is interesting to see the adversarial behavior between the pseudo and the reconstruction loss. In fact, without the reconstruction loss, the pseudo loss would not have any gradient at the beginning (if the pseudo labels are not thresholded), since the pseudo label is produced by the teacher, which at the beginning of the DA is the same as the student. Would a knowledge distillation type of approach, or injection of noise to the student parameters at t=0, reduce the need for the reconstruction loss? Any thought or intuition on why the student=teacher approach should be the to go solution as opposed to a smaller net?

- My understanding is that images are cropped to the largest prostate available in the datasets, does this make it necessary to always have availability of ground truth masks at test time? How would you address the problem when there is no ground truth mask. And also how would you deal with cases where the organ is larger than the cropping size that was used in training?

- In page 6, authors mention confidence with respect to Dice loss. In certain scenarios having a confident model can be undesirable, while I think that the benefit of Dice is mostly related to the ability to better handle class unbalances as opposed to pure CE for instance. Why would you like to make the model to be more confident in your specific case? Have you tried training with CE? And if so, did you only noticed performance changes, or also the confidence behavior was altered?

- Also, what do you mean by background in “not break away from the background image.”

- I suspect that at the very end of page 8, there is a typo when stating results from “five” public datasets.


**Final Rating After The Rebuttal:**

4: Weak Accept

**Justification Of The Final Rating:**

I appreciate the authors clarifications. My initial understanding was that the masks were thresholded before computing the DSC - therefore causing no gradients at the beginning of training when the student is initialized using the teacher's parameter; that led to my comment.

Considering the comprehensive response which the authors have provided also to the other reviewers, I would like to upgrade my rating to weak accept, I think this paper could be of interest for the audience at MIDL.

**Paper Type:**

both

**Questions To Address In The Rebuttal:**

- I would like the authors to expand the related work section. I understand there are page limits, but since there are a few appendix sections, perhaps related work could be expanded in the appendix.

- I would also appreciate some genuine self-criticism about the limitations of this method.

- Usually in teacher – student, the student is trained on the output of the teacher (this constitute the pseudo loss), and because the teacher’s output is a probability map rather than one hot encoded, the student can learn to be less overconfident (like in label smoothing), and achieve teacher – if not better – performance. Here authors have used the teacher to initialize the student, by copy the parameters. I would like the authors to comment on whether they have tried to do knowledge distillation as part of the study.

The code could be made available in full spirit with an open source community, where the medical imaging field could advance together – also 3 out of 4 datasets are publicly available.





**Special Issue:**

no

---

### Official Review · Reviewer_bxhh · 2022-01-24

**Confidence:** 4
**Preliminary Rating:** 4
**Recommendation:** Poster

**Summary:**

This paper proposes a novel approach for the task of unsupervised domain adaptation for medical image segmentation. The key idea is to learn a shape distribution from the source domain, and then to guide the segmentation model adaptation in target domain. The proposed method is a teacher-student pipeline, consisting of a teacher segmentation module, a student segmentation module, and a VAE shape reconstruction module. The experimental comparisons are conducted on three benchmarks, verifying the efficacy of the proposed method. The key idea is interesting and well-motivated. The proposed pipeline is effective. Some discussion and comparisons are suggested in weaknesses.

**Strengths:**

* This paper is overall well-written and easy to follow.
* The key idea of utilizing a VAE to capture organ shape distribution to guide the learning of student model is interesting and well-motivated.
* The proposed pipeline is reasonable and supported with ablative experiments.
* Experimental comparisons are shown on three target datasets.

**Weaknesses:**

* The idea of exploring organ shape information in situations with limited training data has been explored before, e.g., Qian He, Shuailin Li, and Xuming He. "Weakly Supervised Volumetric Segmentation via Self-taught Shape Denoising Model." MIDL 2021. There should be discussion on these two papers.
* How does the model perform if the teacher model is iteratively updated with the student model? The initial pseudo labels can potentially include severe noise.
* How is the quality of the reconstructed prediction masks from VAE, compared to prediction masks from student segmentation model?
* In Section 4.3., the explanation for methods "Pseudo Label" and "Discriminator" is not very clear. It is confusing how these two methods are implemented.

**Deanonymize Review:**

no

**Detailed Comments:**

* In Line 3, Paragraph 3 of Section 4.2., "0.42 Dice score" seems to be "0.042 Dice score".
* In the last sentence of Section 5., are there "five public datasets" in this paper? There seems only to be three target datasets and one source dataset.

**Final Rating After The Rebuttal:**

4: Weak Accept

**Justification Of The Final Rating:**

The authors have addressed most of my concerns. While I am still curious about the quality of the reconstructed prediction masks from VAE, and I think it would be better to include quantitative comparison to prediction masks from student segmentation model. The main concern is that the reconstructed masks could potentially be misaligned and thus misguide the training, even if with a cleaner mask prediction.

**Paper Type:**

methodological development

**Questions To Address In The Rebuttal:**

The main concerns are explained in the "Weaknesses" above. I'd like the authors to address more discussion on other related works utilizing shape prior, and to show some additional experimental analysis to verify the current method design.

**Special Issue:**

no

---

### Meta-Review · Area_Chair_KdEQ · 2022-02-14

**Recommendation:** Accept (Poster)
**Confidence:** 5

**Metareview:**

This paper proposes to improve medical image segmentation via leveraging shape modeling with a variational autoencoder (VAE). They also address unsupervised domain adaptation with a teacher-student pipeline, using a dual-loss function. They additionally implement test-time training to further improve the result. Pancreas segmentation was targeted in this paper, based on 4 datasets coming from various origins.

Strengths:
- This paper is well written. The idea of using shape via a VAE in medical image segmentation is fully relevant and of broad interest.
- The overall technical aspects are sound; presented results, including ablation studies, are convincing.

Weaknesses:
- Some reviewers were missing a discussion on related previous works, and on the work limitations.
- Some reviewers wanted to know more about the behavior of the method in certain cases.
- One reviewer was missing additional experiments, in different ablative settings.

The authors were able to provide answers to the reviewers query, including new experimental results. Thanks to this upgrade, and following the reviewers updated scores, I believe the paper will be in good shape to be accepted at MIDL.

---

### Decision · Program_Chairs · 2022-02-28

Accept